# Characterization of Systemic Disease Development and Paw Inflammation in a Susceptible Mouse Model of Mayaro Virus Infection and Validation Using X-ray Synchrotron Microtomography

**DOI:** 10.3390/ijms24054799

**Published:** 2023-03-02

**Authors:** Ana Carolina de Carvalho, Carlos Sato B. Dias, Laís D. Coimbra, Rebeca P. F. Rocha, Alexandre Borin, Marina A. Fontoura, Murilo Carvalho, Paul Proost, Maurício L. Nogueira, Sílvio R. Consonni, Renata Sesti-Costa, Rafael Elias Marques

**Affiliations:** 1Brazilian National Biosciences Laboratory—LNBio, Brazilian Center for Research in Energy and Materials—CNPEM, R. Giuseppe Máximo Scolfaro, 10000-Bosque das Palmeiras, Campinas 13083-100, Brazil; 2Department of Genetics, Evolution, Microbiology and Immunology, Institute of Biology, University of Campinas (UNICAMP), Rua Bertrand Russel, Campinas 13083-970, Brazil; 3Laboratory of Molecular Immunology, Department of Microbiology, Immunology and Transplantation, Rega Institute, Katholieke Universiteit Leuven (KU Leuven), Herestraat 49 Box 1042, 3000 Leuven, Belgium; 4Institut Für Photonenforschung und Synchrotronstrahlung (IPS), Karlsruher Institut Für Technologie (KIT), Hermann-von-Helmholtz-Platz 1, Eggenstein-Leopoldshafen, 76344 Karlsruhe, Germany; 5Department of Biochemistry and Tissue Biology, Institute of Biology, University of Campinas (UNICAMP), Rua Monteiro Lobato, s/n, Campinas 13083-970, Brazil; 6Brazilian Synchrotron Light Laboratory—LNLS, Brazilian Center for Research in Energy and Materials—CNPEM, R. Giuseppe Máximo Scolfaro, 10000-Bosque das Palmeiras, Campinas 13083-100, Brazil; 7Laboratório de Pesquisas em Virologia (LPV), São José do Rio Preto Medical School (FAMERP), Av. Brigadeiro Faria Lima, 5416-Vila São Pedro, São José do Rio Preto 15090-000, Brazil

**Keywords:** mayaro virus, alphavirus, innate immunity, inflammation, microtomography, synchrotron technology

## Abstract

Mayaro virus (MAYV) is an emerging arthropod-borne virus endemic in Latin America and the causative agent of arthritogenic febrile disease. Mayaro fever is poorly understood; thus, we established an in vivo model of infection in susceptible type-I interferon receptor-deficient mice (IFNAR−/−) to characterize the disease. MAYV inoculations in the hind paws of IFNAR−/− mice result in visible paw inflammation, evolve into a disseminated infection and involve the activation of immune responses and inflammation. The histological analysis of inflamed paws indicated edema at the dermis and between muscle fibers and ligaments. Paw edema affected multiple tissues and was associated with MAYV replication, the local production of CXCL1 and the recruitment of granulocytes and mononuclear leukocytes to muscle. We developed a semi-automated X-ray microtomography method to visualize both soft tissue and bone, allowing for the quantification of MAYV-induced paw edema in 3D with a voxel size of 69 µm^3^. The results confirmed early edema onset and spreading through multiple tissues in inoculated paws. In conclusion, we detailed features of MAYV-induced systemic disease and the manifestation of paw edema in a mouse model extensively used to study infection with alphaviruses. The participation of lymphocytes and neutrophils and expression of CXCL1 are key features in both systemic and local manifestations of MAYV disease.

## 1. Introduction

Mayaro virus (MAYV) is a mosquito-borne virus of the Togaviridae family, Alphavirus genus, with other important human pathogens such as Chikungunya, Ross River, Sindbis and Venezuelan Equine Encephalitis viruses [1]. The MAYV genome is a positive-sense, single-stranded RNA of approximately 11.5 kb comprising four nonstructural proteins (nsPs) involved in viral replication and five structural proteins that participate in the generation and maturation of viral particles [2,3]. First isolated in Trinidad in 1954, MAYV has been detected in several countries in South and Central America. Human cases of Mayaro Fever were reported in Brazil, Bolivia, Costa Rica, French Guiana, Guatemala, Haiti, Panama, Peru, Suriname and Venezuela [4,5]. Although Mayaro Fever occurrence is largely underreported or misdiagnosed as other arboviral diseases, recent changes in MAYV distribution in Brazil indicate that Mayaro Fever cases may be increasing in number and size [6,7]. There are no specific treatments or licensed vaccines against MAYV infections, despite several pre-clinical studies and ongoing phase 2/3 vaccine clinical trials [5,8]. Mayaro Fever consists of a mild, self-limited febrile disease with symptoms such as headache, skin rash, myalgia and arthralgia, but it may evolve into a long-lasting arthritogenic disease with joint inflammation and intense pain, leading to significant morbidity. The inflammasome and neutrophil extracellular traps were shown to participate in alphaviral arthritogenic diseases, but the pathogenesis of Mayaro Fever remains incompletely understood [9,10]. The currently used techniques to characterize and study Mayaro Fever often result in sample destruction or consumption. The use of non-destructive advanced imaging techniques may represent an important advantage, allowing multiple analyses in tissues that are hard to obtain, such as from joints.

X-ray tomography is a well-established technique used in medical radiography to view tissues in a living body using X-rays with a typical resolution of 1 to 2 mm. The achievement of X-ray tomography was recognized in the Nobel prize in Physiology or Medicine in 1979, and since then, it has been widely used in biological and material sciences for non-invasive cross-sectional imaging. Flannery and colleagues developed microtomography in 1987, producing three-dimensional (3D) images of internal structures of small samples with a micrometer resolution. Since then, X-ray microtomography (µCT) can be used to non-destructively create 3D images of internal sections of a sample with a resolution limit comparable to that of a light microscope. 

In this study, we established a mouse model of MAYV infection in susceptible type I interferon receptor-deficient (IFNAR−/−) mice to characterize disease development and paw inflammation. We correlated the onset of edema with other aspects of inflammatory responses induced by the MAYV infection, such as the production of proinflammatory cytokines and leukocyte recruitment. Using a semi-automated phase–contrast X-ray microtomography performed at a synchrotron source, we quantified and characterized MAYV-induced paw edema in soft tissues with precision and high reproducibility. Detailed features of MAYV-induced paw inflammation observed using microtomography should be useful in studying the pathogenesis of other viral diseases in which inflammation is a critical feature.

## 2. Results

### 2.1. Footpad Inoculation of MAYV in Immunodeficient IFNAR−/− Mice Results in Systemic Infection and Disease

We established a susceptible mouse model of the MAYV infection, inoculating adult IFNAR−/− mice with MAYV strain IQT 4235, henceforth referred to as MAYV. To observe aspects of severe disease development, we injected the mice with a high viral inoculum (10^5^ MAYV plaque forming units (PFU)) using different routes of inoculation: intra-articular in the left hind knee, subcutaneous, intraperitoneal or intraplantar also in the left hind limb (Figure 1A). MAYV inoculation by all tested routes resulted in disease development, as the infected mice showed ruffled fur, conjunctivitis, hunched posture and reduced exploratory behavior and movement. Weight loss was also observed in separate experiments in the mice inoculated with 10^2^ or 10^3^ PFU (Appendix A). All the MAYV-infected mice died by day four post-infection (p.i.), while the non-infected (NI) mice remained healthy (Figure 1A). Notably, intraplantar inoculation of MAYV in the left back footpad led to signs of inflammation, with redness and swelling appearing throughout the course of infection, in accordance with previous reports [9,11,12]. Since the intraplantar route of MAYV recapitulates aspects of virus transmission and clinical signs of disease in humans [2,5], we chose 10^2^ PFU intraplantar inoculation for the subsequent experiments. Inoculum-response experiments were performed with MAYV inocula ranging from 10^0^ to 10^3^ PFU per mouse, administered in a total volume of 20 μL. The mock-infected mice received a similar volume of PBS vehicle as a negative control (Figure 1B). All the MAYV inocula caused disease signs up to day 4 p.i., as observed in the previous experiment. The mice inoculated with 10^3^ or 10^2^ died on day 4 p.i., while mice inoculated with 10^1^ or 10^0^ PFU survived up to day 6 p.i. Signs of paw inflammation were observed in all the infected mice, though higher inocula led to earlier signs of inflammation (Figure 1C–F). At day 4 p.i., PBS-inoculated paws in mock-infected mice had no apparent signs of inflammation (Figure 1C,E), while MAYV-infected paws inoculated with 10^2^ PFU were swollen and reddish (Figure 1D,F).

To investigate the course of MAYV infection in this model, we inoculated IFNAR−/− mice with 10^2^ PFU of MAYV or vehicle. Samples of serum, liver, spleen and brain were collected on days 1, 2 and 3 p.i. to assess viral load by a plaque assay (Figure 1G–J). Within 1 day p.i., the infectious MAYV particles were detected in most tissues tested, indicating a fast viral dissemination and the onset of viremia (Figure 1G). At day 1 p.i., we observed brain samples that were either negative for the presence of MAYV or had a reduced viral load in comparison to other organs, indicating that MAYV takes longer to disseminate to the brain in comparison to the spleen and liver (Figure 1H–J). At days 2 and 3 p.i., MAYV replication reaches a plateau in all evaluated organs, with viral loads reaching up to 10^8^ PFU per mL of sample. Organ samples collected from mock-infected mice were negative for MAYV. Levels of proinflammatory chemokines and cytokines CXCL1, CCL5, IFN-γ, IL-1β, CXCL10, IL-17, TNF-α and IL-6 were also assessed in the brain, spleen and liver of MAYV-infected mice throughout the infection (Figure 1K–N and Appendix A). We observed a significant increase in the levels of proinflammatory mediators CXCL1, CCL5, IFN-γ and IL-1β on day 3 p.i. in the spleen of MAYV-infected mice, in comparison to mock-infected controls (Figure 1K–N). While the chemokine CXCL1 was upregulated in all organs tested at day 3 p.i. (Figure 1K and Appendix A), CCL5, IFN-γ and IL-1β were upregulated only in spleen samples from MAYV-infected mice (Figure 1L–N) and remained at baseline levels or were undetected in liver and brain samples from infected mice (Appendix A–H). Levels of CXCL10, IL-17, TNF-α and IL-6 were undetectable in tissue samples from both mock- and MAYV-infected mice.

In summary, we observed that MAYV inoculation in IFNAR−/− mice led to an invariably fatal disease, as expected for this immunodeficient model. The intraplantar inoculation of MAYV leads to noticeable paw inflammation, which is concomitant with rapid viral dissemination and induction of proinflammatory mediators in target tissues, including the neutrophil chemoattractant chemokine CXCL1.

### 2.2. Inoculation of MAYV in Immunodeficient IFNAR−/− Mice Results in Lymphopenia and in Significant Reduction of Leukocyte Populations in the Spleen

We assessed circulating and splenic leukocyte populations in MAYV-infected mice using total and differential counting and flow cytometry. Blood was collected from MAYV- and mock-infected mice at days 1, 2 and 3 p.i. The total and differential counting of leukocytes in blood smears indicated that MAYV-infected mice developed leukopenia already at day 1 p.i., which aggravated gradually to day 3 p.i. (Figure 2A). MAYV-induced leukopenia is mostly due to lymphopenia, as lymphocyte counts in infected mice at day 3 p.i. were almost 70% lower in comparison to mock-infected controls, while numbers of polymorphonuclear (PMN) and mononuclear cells were similar between groups. 

Spleen samples collected from mock- and MAYV-infected mice at day 3 p.i. were processed and analyzed using flow cytometry to assess both the numbers and activation status of leukocyte populations (Figure 2B–Q). Spleen weight varied between mock- and MAYV-infected mice, in which spleens from infected mice were heavier (Figure 2B). Conversely, the number of splenocytes recovered from MAYV-infected mice was significantly less than those recovered from mock-infected mice (Figure 2C). T CD4+ lymphocytes (CD3+CD4+ cells) were the most abundant leukocyte population in the spleen of both mock and MAYV-infected IFNAR−/− mice and were found to be reduced in the MAYV-infected group in comparison to control mice (Figure 2D). T CD8+ lymphocytes (CD3+CD8+ cells) and CD11b- dendritic cells (DCs) (CD11c+CD11b- cells) were also found to be reduced in the spleens of infected mice (Figure 2E,G). The number of macrophages (F4/80+CD11b+ cells) was strikingly reduced in the spleens of MAYV-infected mice, with an average 20-fold decrease in comparison to mock-injected mice (Figure 2H). The number of CD11b+ DCs (CD11c+CD11b+ cells) and neutrophils (Ly6G+CD11b+ cells) were similar between the mock- and MAYV-infected mice (Figure 2F,I). The populations of B lymphocytes and NK cells were not assessed. 

Although splenic T CD4+ and T CD8+ lymphocyte populations were found to be reduced in the MAYV-infected mice in comparison to the mock-infected mice, we observed that these populations were more activated, expressing higher levels of the activation marker CD69 (Figure 2J–M). Neutrophils (Figure 2N,O) and both CD11b+ and CD11b- conventional DCs (Appendix A) expressed higher levels of the selectin CD62L, involved in leukocyte recruitment to inflamed tissues. In contrast, the depleted macrophage population observed in the spleens of MAYV-infected mice expressed less MHC-II, indicating that this leukocyte population was also not activated in the context of the MAYV infection (Figure 2P–Q). the MHC-II downregulation was specific to macrophages, as DCs expressed baseline or upregulated levels of MHC-II in comparison to mock-infected controls (Appendix A). 

In summary, the inoculation of MAYV in the paws of IFNAR−/− mice leading to disseminated infection and disease also results in significant changes in leukocyte populations in the blood and spleen. These changes include dramatic reductions in leukocyte populations coupled with increased activation phenotypes, notably in lymphocytes and neutrophils.

### 2.3. Paw Inflammation in MAYV-Infected Mice Involves Edema in Multiple Tissues, Local Production of Cytokines, Leukocyte Recruitment and Infection

Histopathological analysis was performed to further characterize MAYV-induced inflammation in mice paws (Figure 3). An analysis of 5 μm-thick longitudinal footpad sections stained with hematoxylin–eosin confirmed that intraplantar MAYV infection leads to accentuated edema and the disruption of tissue architecture in mouse paws, while the mock-infected paws presented normal tissue architecture (Figure 3A–D). Micrographs from the dermis (Figure 3E–H) and from muscles and fascia (Figure 3I–L) were taken from paws of mock-infected mice and MAYV-infected mice at days 1, 2 and 3 p.i. and indicated that tissue alterations aggravate along the evaluated period. Although we visually confirmed MAYV-induced paw inflammation only at day 3 p.i., the histological analysis indicated soft tissue edema on day 2 p.i. Leukocyte infiltration was observed in the skin of infected mice only at day 3 p.i. and was characterized by the presence of granulocytes (Figure 3E–H). Leukocyte infiltration was more intense in the muscles and fascia, where both granulocytes and mononuclear cells were present in tissues from day 1 p.i., remaining consistent until day 3 p.i. (Figure 3I–L). Vascularity and diapedesis in the muscles and fascia took place on day 1 p.i., preceding the leukocyte recruitment in the MAYV-infected paws. Our results from soft tissue analysis are summarized in the semi-quantitative histological scores in Figure 3M,N. We observed no signs of erosion, leukocyte infiltration or synovial hyperplasia in bones and joints of mock- or MAYV-infected mice, indicating that the histoarchitecture of paw bones and joints was not affected by MAYV in the evaluated period (Figure 3O). 

Moreover, the quantification of viral load in paw soft tissues and popliteal lymph nodes by plaque assay confirmed that MAYV is present in these structures in infected mice on day 3 p.i. at a high viral load (above 10^7^ PFU per mL of tissue homogenate), comparable to viral loads observed in serum, liver, spleen and brain samples from infected mice (Figure 3P,Q). No virus was retrieved from the paws of control mice. The MAYV presence in inflamed paw soft tissue was accompanied by increased levels of CXCL1 on days 2 and 3 p.i. when compared to mock-infected controls (Figure 3R). The upregulation of this proinflammatory chemokine in the paws of the MAYV-infected mice is concomitant with the infiltration of leukocytes (Figure 3M,N) and with CXCL1 upregulation in other organs (Figure 1K and Appendix A). 

In summary, the intraplantar MAYV infection in IFNAR−/− mice causes major changes to soft tissues in paws, including edema, the disruption of tissue architecture, increased vascularity/diapedesis and leukocyte infiltration. The local production of the proinflammatory mediator CXCL1 is concomitant to viral infection and in line with the presence of granulocytes in dermis, muscle and fascia.

### 2.4. An X-ray Synchrotron Microtomography Method to Assess Paw Inflammation in MAYV-Infected Mice in 3D

MAYV-induced paw inflammation is an obvious but rather complex phenomenon in susceptible IFNAR−/− mice. To better understand edema development and distribution, we developed a semi-automated microtomography technique to quantify paw edema, with precision, in 3D (Appendix A). A single tomogram is composed of 1024 projections and takes approximately 20 min to be completed, generating 8 GB of data. Since our previous results indicated that MAYV-induced paw inflammation took place in the dermis, muscle and fascia, our method was designed to observe soft tissues in addition to bone, which is easily contrasted using X-rays. By exploiting the phase–contrast present on the projections, we were able to reconstruct a 3D image with high contrast regarding the skin and fur and a low contrast image regarding internal structures such as muscle, blood vessels, fat, fascia, ligaments and edema, that could be virtually sliced for visualization. A qualitative description of the edema could be made from cross-sectional virtual slices, indicating that the edema is located mostly in and below the dermis, confirming our previous result using histology and a semi-quantitative pathology score (Appendix A; Figure 3). We could also observe edema in the distal regions of fingers, indicating that the entire paw is affected (Appendix A).

We observed the contrast between regions of interest in µCTs of mouse paws. Soft tissues could be separated from empty tomography space due to phase contrast in the paw contour (skin and hair), and bone was easily visualized due to mineral density. Such differences allowed for the implementation of a semiautomatic segmentation procedure, as manual segmentation is labor-intensive and error-prone. Our strategy involved the use of Level-set methods for soft tissue segmentation and thresholding for hard tissue segmentation. For soft tissue, the segmentation strategy relied on a contour refining algorithm able to find the entire volume of the paw within the whole tomographic volume (2048 × 2048 × 2048 pixels). For bone, a regular contrast threshold analysis was performed, in which each pixel in an image was replaced by a pixel in greyscale, size and shape compatible with X-ray hard biological tissue. Pixels matching the characteristics of bone were segmented accordingly and allowed for the precise segmentation of entire paw bones and joints. The algorithms adopted for the segmentation of both tissues required an initial optimization of multiple parameters, and the same protocol was run on each paw separately, with an occasional inspection to ensure a successful segmentation. The resulting segmented volumes could then be used for quantitative volume visualization and calculation (Appendix A).

We initiated our microtomographic analysis using the inoculated paw from mock- and MAYV-infected mice at days 1, 2 and 3 p.i. (Figure 4A). Our results indicated that the soft tissue volume increases gradually during day 1 and 2 p.i. but rises significantly at day 3 p.i., with approximately a 50% increase in volume in comparison to the mock-infected paws (Figure 4A). Next, we tested how robust our microtomography technique would be and whether we could quantify less intensive MAYV-induced edema. The IFNAR−/− mice were inoculated with saline (mock) or 10^1^ MAYV PFU and euthanized at day 4 p.i. for the collection of both inoculated (ipsilateral) and non-inoculated (contralateral) hind paws (Figure 4B,C). We generated 72 µCTs in 48h at the synchrotron source IMX beamline generating a total of 618 GB of data. We used strict parameters for data acquisition to minimize variations between paws and between segments of the same paw. The µCTs were reconstructed and combined in a total of 24 paws (three tomograms per paw covering a larger field-of-view (FOV)). The µCTs of complete paws were segmented using the same segmentation procedure and resulted in consistent binary masks for both soft and hard tissue. The quantitative values for volume were obtained by counting the number of voxels on each mask and accounting for the voxel size. A comparison of paw volumes between ipsilateral mock- and MAYV-infected mice indicated an average 21% increase in soft tissue volume in infected paws, from 47 to 57 mm^3^ (Figure 4B). The contralateral paws from mock- and MAYV-infected mice had similar soft tissue volumes, indicating that non-inoculated paws did not develop edema (Figure 4C). Moreover, we did not observe differences in bone volume between any of the evaluated groups, confirming that paw bones are largely unaffected in this model of MAYV infection. Segmented volumes of mock- (Figure 4D) and MAYV-infected ipsilateral paws (Figure 4E) indicated that mock-infected paws present a normal tissue structure and volume, while infected paws swell in both dorsal and ventral areas, exemplified by the protruding walking pads. 

We conclude that this semi-automated microtomography method is suitable for studies of tissue architecture in 3D, allowing for non-destructive, precise and quantitative measurements of volumes of both soft and hard tissues and recapitulating results from the currently established methods, such as histology. The MAYV-induced inflammation permeated multiple tissues in inoculated paws and manifested mostly as edema, while bones and non-inoculated paws remained unaffected by MAYV in this model.

## 3. Discussion

In vivo models of the alphavirus infection are important tools for investigating the mechanisms of alphavirus-induced disease development and resistance, as well as for assessing the efficacy and safety of candidate treatments [11,13]. Adult immunocompetent wild-type (WT) mice are resistant to MAYV inoculation, while young mice or mice deficient in Type-I IFN responses are susceptible and develop disease [8,14]. Our objective was to better understand the main features of MAYV-induced disease in vivo and thus establish the susceptible mouse model of MAYV infection using IFNAR−/− mice. MAYV infection promoted an acute and lethal disease, with manifestations that corroborate previous studies [14,15]. The footpad injection of IFNAR−/− mice with MAYV inocula as low as 1 PFU led to the development of disease signs and death up to 6 days p.i. Paw inflammation was observed from day 1 p.i. and was characterized by edema, the disruption of tissue architecture and the infiltration of leukocytes. Eye irritation was observed in infected mice in all experiments, corroborating a study by Santos and colleagues [12], and was predictive of death in this model. Ocular symptoms, although commonly described in human MAYV cases, are overlooked in animal models. Ocular manifestations take place in human infections with other alphaviruses (Chikungunya, Ross River, Eastern, Western and Venezuelan Equine Encephalitis viruses) and might indicate the involvement of the central nervous system (CNS) [16,17,18]. MAYV, Chikungunya virus (CHIKV) and other alphaviruses of the Semliki Forest complex cause diseases with similar clinical manifestations in humans [1], suggesting that these viruses may share mechanisms of pathogenicity. We expect our findings on MAYV-induced paw inflammation and systemic disease to be translatable to other models of alphaviral disease, including the application of soft-tissue µCT in research.

We observed elevated viremia and the presence of MAYV in the spleen, liver and brain of MAYV-infected mice one day after intraplantar infection (Figure 1). We interpret the accumulation of MAYV in the brains of IFNAR−/− as a possible limitation of our in vivo infection model, arising from the severe susceptibility of the IFNAR−/− mouse strain to alphavirus infection due to an abrogated type-I interferon response. Such conditions result in the widespread dissemination of the virus to mouse tissues, some of which may not recapitulate disease in humans. Changes in proinflammatory cytokine and chemokine levels, leukocyte populations and histological scores, observed later in infection, were concomitant with disease signs (Figure 3). Chemokine and cytokine expression is concentrated mostly in the spleen of infected mice. The expression of CXCL1, CCL5, IFN-γ and IL-1β were found to be increased in the spleen, in accordance with previous reports for IFNAR−/− mice MAYV models [11,14]. However, cytokines such as IL-6 and TNF-α that are typically associated with inflammatory responses were absent in all the organs and time points evaluated. The expression of CXCL10, a proinflammatory chemokine induced by IFN-γ and important in antiviral responses [19], was absent in the brain, spleen and liver of infected mice, though IFN-γ was upregulated in the spleen. Collectively, these observations suggest that expression of proinflammatory mediators such as IL-6, TNF-α and CXCL10 may depend on intact type-I interferon responses and that MAYV-induced inflammation and disease are not associated with these mediators in this model of infection.

In addition to the inoculated paw, the spleen seems to be most affected by MAYV infection in this model. Although spleens of MAYV-infected mice are heavier than spleens from mock-infected mice, they contain, on average, less than half of the cells, indicating reduced cellularity and edema in the organ. The total numbers of CD4+ T lymphocytes (T CD4+), T CD8+, macrophages and CD11b- dendritic cells were reduced, while neutrophils and CD11b+ dendritic cells remained unchanged. The reduction in the absolute number of macrophages (more than 20-fold reduction) and T CD4+ lymphocytes (2-fold reduction), the latter being the most abundant leukocyte population in the spleen, are likely to account for the total reduction in spleen cellularity observed in MAYV-infected mice. MAYV replicates in RAW 264.7 and J774 macrophage cell lines and primary cultures of peritoneal and bone marrow-derived macrophages. It leads to the production of reactive oxygen species, secretion of proinflammatory chemokines, apoptosis and cell death [9,20]. Thus, we speculate that MAYV infection in splenic macrophages is leading to cell death in vivo, ultimately resulting in the reduction of this population and spleen inflammation in infected mice. Moreover, the loss of functional splenic macrophages may explain the absence of TNF-α expression in the spleen, as macrophages are important sources of this cytokine [21,22]. 

Our results suggest that neutrophils participate in the inflammatory response to MAYV infection in both the paw and the spleen. The neutrophil chemoattractant CXCL1 was upregulated in the MAYV-infected mice in all organs tested, including the inflamed paw. De Castro-Jorge and colleagues [9] demonstrated that neutrophils are recruited to MAYV-infected paws in WT immunocompetent mice, which corroborates our observation of PMN recruitment to MAYV-infected muscle and ligaments in susceptible IFNAR−/− mice and indicates that this process is independent of intact type I IFN responses (Figure 3). Interestingly, the neutrophil population in the spleen, although similar in number between infected and non-infected mice, was activated in infected mice and expressed higher levels of CD62L (Figure 2). CD62L, also known as L-selectin, is an adhesion molecule involved in the recruitment of leukocytes to inflamed tissues, which is typically shed from the surface of neutrophils upon integrin activation and after transmigration from the bloodstream to the tissue [23]. Thus, activated neutrophils in target tissues would have lower levels of CD62L on the surface, as assessed by conventional flow cytometry. Neutrophils in the spleen of MAYV-infected mice express higher levels of CD62L than uninfected control mice, indicating an abnormal activation status. The implication of higher CD62L expression on neutrophils during alphavirus diseases is unknown, but previous studies from our group with bronchoalveolar lavage samples from severe COVID-19 infected human patients in ICU also presented abnormal activation of neutrophils, pointing towards a possible role for neutrophils contributing to the severity of inflammation upon viral infections [24].

Although three-dimensional visualization of mice anatomy was previously demonstrated, imaging was often restricted to bone or other tissues with high mineral content or required extensive sample preparation with the use of toxic, corrosive reagents for tissue clearance or enhancement prior to measurements [25,26]. We established the first µCT technique that also allows imaging of soft tissue to visualize unappreciated features of MAYV infection and disease without contrasting agents. The 2D virtual slices obtained from the reconstructed volumes corroborate results obtained by conventional histopathology assays, with the advantage of visualizing multiple axes or points of view without the need to prepare new samples from the start, a requirement for histological methods [27,28]. Using soft-tissue µCT, we were able to confirm previous results on MAYV-induced edema produced using visual observation and histology and added detailed information on edema localization and tissue architecture. A main motivation for the use of soft tissue µCT is the possibility of obtaining quantitative measurements of tissue volume with high precision. Each pixel in 2D virtual slices of mouse paws corresponds to 16.9 μm^2^, and each voxel in 3D volumes to 69 μm^3^, which represent a significant increase in precision and sensitivity in comparison to instruments such as a paquimeter or plethysmometer, used to assess edema in biological samples. Although the manual segmentation of volumes can be more precise, the process is time-consuming and has low reproducibility and scalability. Thus, to overcome these limitations and allow for comparisons between individuals and experimental groups, we limited manual intervention in the soft tissue µCT analysis process to a minimum: sample positioning according to an anatomical region (the joint between distal and intermediate phalanges in the middle finger). Soft tissue µCT should be useful not only for the study of edema and inflammation but it could be adapted for imaging different target tissues or anatomical structures. Different sample preparation strategies and tissue-specific staining using X-rays would allow the segmentation of multiple soft tissues and structures [29,30].

The scalability of segmentation is a reflection not only of the chosen segmentation algorithms but also due to the robustness of the µCT measurement and reconstruction achieved by the rigorous control of all the imaging and reconstruction parameters described above. Moreover, when combined with labeling techniques for the detection of specific molecules, such as immunogold or in situ hybridization, soft tissue µCT becomes a valuable resource for the characterization of cell types and molecules involved in many aspects of the inflammatory process. Such a combination would allow virtual histology analyses and the precise quantification of healthy and damaged tissues.

In conclusion, this in vivo model of MAYV infection in IFNAR−/− mice allowed observation of important features of disease and inflammation induced by MAYV and is corroborated by previous studies [8,9,11,12,14,15]. We are the second group to describe eye irritation in mice, a common manifestation in Mayaro Fever patients that is overlooked in animal studies. The systemic spread of MAYV with high viremia and the upregulated expression of proinflammatory chemokines and cytokines was observed. The absence of CXCL10, IL-1β and TNF-α expression may be explained by the abrogated type-I interferon response and a sharp reduction in splenic macrophages, one of the main sources of TNF-α. Neutrophils also participate in MAYV pathogenesis in this model, and an abnormal activation status was observed in the spleen; however, their participation in disease pathogenesis needs further studies. Finally, we established a μCT technique that allows visualization of both bones and soft tissue in high definition, enabling the exploration of unappreciated features of MAYV-induced disease. As a perspective, the combination of paw μCT with staining techniques based on heavy metal labeling should further increase the scope of the application of μCT in biomedical research.

## 4. Materials and Methods

### 4.1. Viral Stock

The MAYV strain IQT 4235 (GenBank accession number MK070491.1), originally isolated from a symptomatic patient in the Peruvian Amazon, was provided by Dr. Scott Weaver (UTMB, Galveston, TX, USA). Viral stocks used in the experiments were generated by passages in mycoplasma-free Vero CCL81 cells (ATCC, Manassas, VA, USA), cultivated in T75 culture flasks with DMEM supplemented with 10% *v/v* FBS and 1% *v/v* penicillin–streptomycin (Gibco, Billings, MT, USA) and incubated at 37 °C in a 5% *v/v* CO_2_ atmosphere. Culture supernatant was harvested on observation of a 50% cytopathic effect, clarified (3000 rpm, 8 min), aliquoted and stored at −80 °C until use. Viral titers of each stock were determined by a plaque assay.

### 4.2. Mice

Adult (8–12 weeks old) male and female IFNAR−/− mice in the SV129 background, which are deficient for the type I IFN receptor, were bred in the Animal Facility of CNPEM. Up to five mice were allocated in cages with access to water and food ad libitum. The Animal facility operates in specific pathogen-free conditions with controlled temperature and photoperiod (21 °C, 12 h/12 h dark/light cycle). All experiments were performed under the guidelines and approval of the CNPEM Committee on Ethics and Use of Animals (CEUA) under protocols 30B and 79.

### 4.3. In Vivo MAYV Infection and Disease Model

IFNAR−/− mice were injected with MAYV IQT 4235 at different inocula (from 10^5^ PFU/animal to 10^0^ PFU/animal) through intra-articular (knee), subcutaneous (back), intraplantar or intraperitoneal routes. Mock-infected controls received the same volume of PBS. In lethality experiments, weight loss and disease signs (changes in behavior and movement, hunched posture, ruffled fur, loss of grip, ocular inflammation) were monitored daily from day 0 post-infection (p.i.) until day 6 p.i. For assessment of viral replication, histopathology, ELISA, flow cytometry and X-ray microtomography experiments, mice were injected with MAYV IQT 4235 at 10^2^ PFU/animal via the intraplantar route, and weight loss and disease signs were monitored daily from day 0 to day 3 p.i. Blood, brain, liver, spleen and hind paws were collected on days 1, 2 and 3 p.i. and prepared according to the subsequent experiment. Mice were euthanized with an overdose of anesthetic (ketamine and xylazine in PBS) with subsequent cervical displacement prior to sample collection.

### 4.4. Viral Load Quantification 

Viral load in all samples and viral stocks were determined through a plaque assay in Vero CCL81 cells. Briefly, organs were weighed and macerated to a 10% *w/v* solution with PBS. Organ suspensions, or liquid samples such as blood plasma or culture supernatant, were serially diluted 10-fold ranging from 10^−1^ to 10^−6^ and incubated with confluent Vero cell monolayers in 24 wells plates, for 1h at 37 °C, 5% CO_2_. After adsorption, supernatant was removed, and 2mL of viscous complete DMEM supplemented with 2% *w/v* carboxymethylcellulose was added to each well and incubated for 4 days. Assay plates were fixed in 10% *w/v* paraformaldehyde and stained with 1% *w/v* methylene blue. Viral titers in samples were determined by counting viral plaques, and results were expressed as PFU/mL of serum/culture supernatant or PFU/mg of organ.

### 4.5. Histopathology 

Mice were euthanized with 1:2 ketamine–xylazine solution (85% *v/v* in PBS) intraperitoneally. Hind paws were collected, fixed in paraformaldehyde 4% *w/v* in PBS and kept in 70% ethanol until paraffin embedding. Sections 5 μm thick were obtained from longitudinal cuts parallel to the footpad, mounted on glass slides and stained with hematoxylin–eosin (H&E). Analysis of edema, vascularity/diapedesis and cell infiltration on observed tissues was made through light microscopy. A histological score was created to assess progression of disease signs at tissue level based on previous work by Ierna et al. [31]. For the skin, muscle and fascia, the following parameters were scored: infiltration of mononuclear cells: none (0); mild (1); moderate (2); severe (3); infiltration of PMN cells: none (0); mild (1); moderate (2); severe (3); Vascularity/Diapedesis: none (0); mild (1); moderate (2); severe (3); Edema: none (0); mild (1); moderate (2); severe (3). For bone: infiltration of mononuclear cells: none (0); mild (1); moderate (2); severe (3); infiltration of PMN cells: none (0); mild (1); moderate (2); severe (3); Erosion: none (0); mild surface (1); moderate (2); extensive full thickness (3); Hyperplasia (synovia): none (0); mild surface (1); moderate (2); extensive full thickness (3).

### 4.6. Quantification of Cytokines and Chemokines

Enzyme-linked immunosorbent assays (ELISA) were used to quantify the levels of the cytokines and chemokines CXCL1, CCL5, IFN-γ, IL-1β, CXCL10, IL-17, TNF-α and IL-6 in samples of mouse spleen, liver and brain. Samples were weighed and homogenized to a 10% *w/v* solution in tissue extraction buffer. Assays were performed using R&D’s DuoSet^®^ ELISA Development Systems kits (R&D Systems, Minneapolis, MN, USA) according to the manufacturer’s instructions. Samples with cytokine levels below the limit of detection of each assay are indicated by N.D. (not detected).

### 4.7. Hematological Parameters 

Peripheral blood from brachial plexus of mice was collected in EDTA-coated tubes on days 1 to 3 p.i. For total leukocyte counting, 10 µL of fresh blood was diluted in 90 µL Turk’s solution for erythrocyte lysis and leukocyte staining. Counting was performed with 10 µL of the Turk-blood dilution into Neubauer chambers under optical microscopy. For differential leukocyte counting, smear slides were made with 10 µL of blood, followed by staining with the Panoptic Fast Stain Kit (Laborclin) according to the manufacturer’s instructions. Stained smear slides were observed in an optical microscope. A minimum of 100 cells were counted per slide and leukocytes were identified and counted as mononuclear or polymorphonuclear cells according to the nuclear morphology. Numbers of mononuclear and polymorphonuclear cells observed were used to calculate the proportion of each population within the total leukocyte counting. Total and differential leukocyte counting results are expressed as 10^4^ cells/mL of blood.

### 4.8. Flow Cytometry 

Mice spleens were collected and macerated in 3.5 mL ice-cold DMEM with 10% *v/v* FBS and centrifuged at 1200 rpm, 4 °C, for 10 min. Erythrocytes were lysed using ACK Lysing Buffer (Gibco) according to the manufacturer protocols. Cells were counted under optical microscopy and 2 × 10^6^ cells per well were seeded in a 96-well plate in 200 μL per well. Single-cell suspensions from the spleen were incubated with 2.5 g/mL Fc block (clone 93, Biolegend, San Diego, CA, USA) to block non-specific binding for 10 min at room temperature.

Cells were stained with monoclonal antibodies anti-CD3 APC (17A2), CD4 PE (GK1.5), CD8 FITC (53-6.7), CD11b PE-Cy7 (M1/70), CD11c APC-Cy7 (N418), CD25 APC-Cy7 (PC61), CD62L FITC (MEL-14), CD69 PE-Cy7 (H1.2F3), F4/80 PE (BM8), Ly6G APC (1A8), IA/IE Percp-Cy5.5 (M5/114.15.2), all from BioLegend (San Diego, CA, USA), for 30 min in FACS Buffer at 4 °C, washed and resuspended in paraformaldehyde 2% before acquisition on the flow cytometer (FACS Canto II; BD Biosciences, San Diego, CA, USA). Data were analyzed using FlowJo software (BD Bioscience). 

### 4.9. Microtomography

The mouse paws were placed inside a pipette tip in an upright position and mounted at the rotation stage. By revolving the sample around a fixed rotation axis by 180 degrees in uniformly spaced angular steps to compose a sinogram that was later computationally transformed into the 3D map of the sample, 1024 X-ray transmission images were acquired. The transmission images were obtained using radiation from the 1.67 T bending magnet of the 1.37 GeV UVX storage ring, filtered by 0.55 mm Si filters. These setups produced a polychromatic beam with peak energy at approximately 13 keV and 50% bandwidth. The radiographs were recorded by an indirect detector system (PCO.2000 14-bit CCD cooled camera) and 2X objective, with a pixel size of 4.11 × 4.11 μm^2^. Considering the detector pixel array size of 2048 × 2048, the field of view (FOV) of the radiographs was 8.4 mm × 8.4 mm. This FOV was sufficient to guarantee compact support for the tomography (horizontal axis) but not for imaging the entire paw in the vertical direction. Thus, for each paw 3 tomograms on 3 different heights were obtained. Data were reconstructed using the standard filtered back projection with parzen window, as implemented at the ASTRA toolbox. Finally, the 3D reconstructions of each height were digitally combined.

### 4.10. Segmentation and Volume Analysis 

The 3D reconstructed data obtained, as mentioned previously, were combined using the Avizo software. Each sample consisted of 3 independent tomograms that were aligned and merged into a single 3D volume. Following, all post-processing was performed using the free Fiji software. Those would consist of data cropping and slice denoising by non-local means. After denoising, volume quantification involved two distinct strategies for both soft-tissue and bone segmentation. 

For the soft-tissue segmentation, the Level Sets algorithm (Level Sets and Fast Marching, https://imagej.net/plugins/level-sets, accessed on 23 August 2022) allowed the segmentation of paw boundary on a 2D reconstructed slice. Briefly, this algorithm iterates a contour line until it hits the object boundary. Some parameters like curvature and convergence will prevent the contour from leaking if there are gaps in the object boundary and were properly adjusted for proper segmentation avoiding most of the paws’ hair without leaking into the paw slice. Finally, as the Level Set algorithm is a 2D segmentation tool, a slice-by-slice strategy with each slice using a dilated initial guess based on the previous slice was implemented. In this sense, this was a semiautomatic segmentation strategy, as the initial slice had to be segmented under supervision for parameter optimization. 

For the bone segmentation, a threshold using the minimum criteria on the entire volume histogram generated a segmentation of almost all the bones. Still, a morphological filter based on a minimum volume and minimum perimeter-to-area ratio excluded any structure that was not the bone tissue. Finally, the morphologically filtered image was used as a seed for the Morphological Reconstruction 3D from MorphlibJ, resulting in the final bone binary mask.

Afterward, the obtained binary masks were inspected, and, if necessary, the segmented regions were manually refined. After segmentation of both soft tissue and bone, the entire 3D model was digitally sectioned at the joint connecting the distal and intermediate phalange in the middle finger. A box of approximately 6 × 6 × 6 mm was used to delimit the paw so that all the samples would account for the same anatomic features. The quantitative analysis of each tissue and paw was performed using the software MorpholibJ 3D analyze.

### 4.11. Statistical Analysis

Viral load, ELISA cytokine levels, flow cytometry and total leukocyte counting data were analyzed using Kruskal–Wallis nonparametric test and Dunn’s multiple comparisons’ post-test. Differential leukocyte counts and segmented volume data were analyzed with ordinary 2-way Analysis of Variance (ANOVA), followed by Tukey’s multiple comparison’s test. Weight loss was analyzed by multiple t-tests corrected for multiple comparisons using the Holm-Sidak method. Survival curves were compared by Mantel–Cox log-rank test. Differences between groups were considered significant if *p* < 0.05. All analyses were performed using the GraphPad Prism 8 software. 

## 5. Conclusions

In summary, our data showed that MAYV footpad inoculation in IFNAR−/− mice lead to fatal disease with widespread virus dissemination and replication, causing drastic reductions in leukocyte populations in blood and spleen and increased activation phenotypes, especially for lymphocytes, macrophages and neutrophils. Major changes in soft tissues were observed, including edema, disruption of tissue architecture, increased vascularity/diapedesis and leukocyte infiltration, and we highlight the increased expression of the neutrophil chemoattractant CXCL1 concomitant to damage and presence of virus in the tissues. Neutrophils seem to play a key role in MAYV-induced inflammation, as they are highly present in muscles and ligaments of MAYV-infected IFNAR−/− mice, suggesting that recruitment of this cell population to inflammation sites is independent of intact type-I IFN responses. Finally, we developed a new technique for 3D visualization of inflamed paw structures and precise volumetric quantification of edema and bone loss that requires no previous treatment of the samples, saving time, funding and reagents. We expect our findings on MAYV-induced paw inflammation and systemic disease to be translatable to other models of alphaviral disease, including the application of soft-tissue µCT in research.

## Figures and Tables

**Figure 1 ijms-24-04799-f001:**
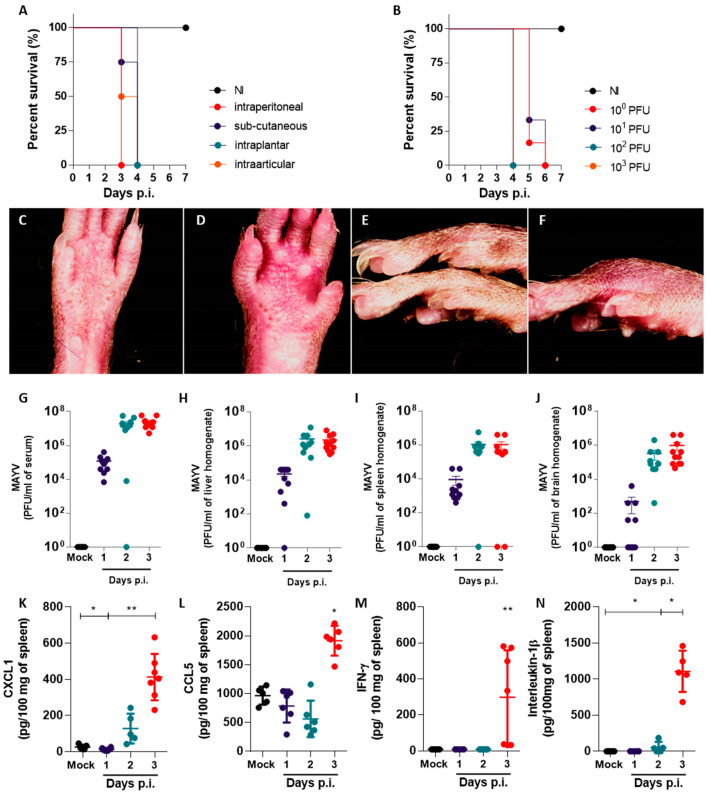
Footpad inoculation of MAYV in immunodeficient IFNAR−/− mice results in systemic infection and disease. (**A**) Survival curves in IFNAR−/− mice infected with 10^5^ PFU by different inoculation routes. (**B**) Survival curves in IFNAR−/− mice infected with different concentrations of MAYV (10^0^, 10^1^, 10^2^, and 10^3^ PFU) via intraplantar inoculation. (**C**–**F**) Representative photographs of mice hind paws injected with (**C**,**E**) PBS vehicle or (**D**,**F**) 10^2^ PFU MAYV at day 3 p.i., showing inflammation signs (redness, swelling). (**G**–**J**) Viral load observed in (**G**) serum, (**H**) liver, (**I**) spleen and (**J**) brain of mock- and MAYV-infected mice at days 1 to 3 p.i. (**K**–**N**) Levels of the chemokines (**K**) CXCL1 and (**L**) CCL5 and the cytokines (**M**) IFN-γ and (**N**) IL-1β observed in the spleen of mock- and MAYV-infected mice at days 1 to 3 p.i. PFU = Plaque Forming Units, p.i. = post-infection. Data presented as mean ± SEM. * *p* < 0.05, ** *p* <= 0.01 See also Appendix A.

**Figure 2 ijms-24-04799-f002:**
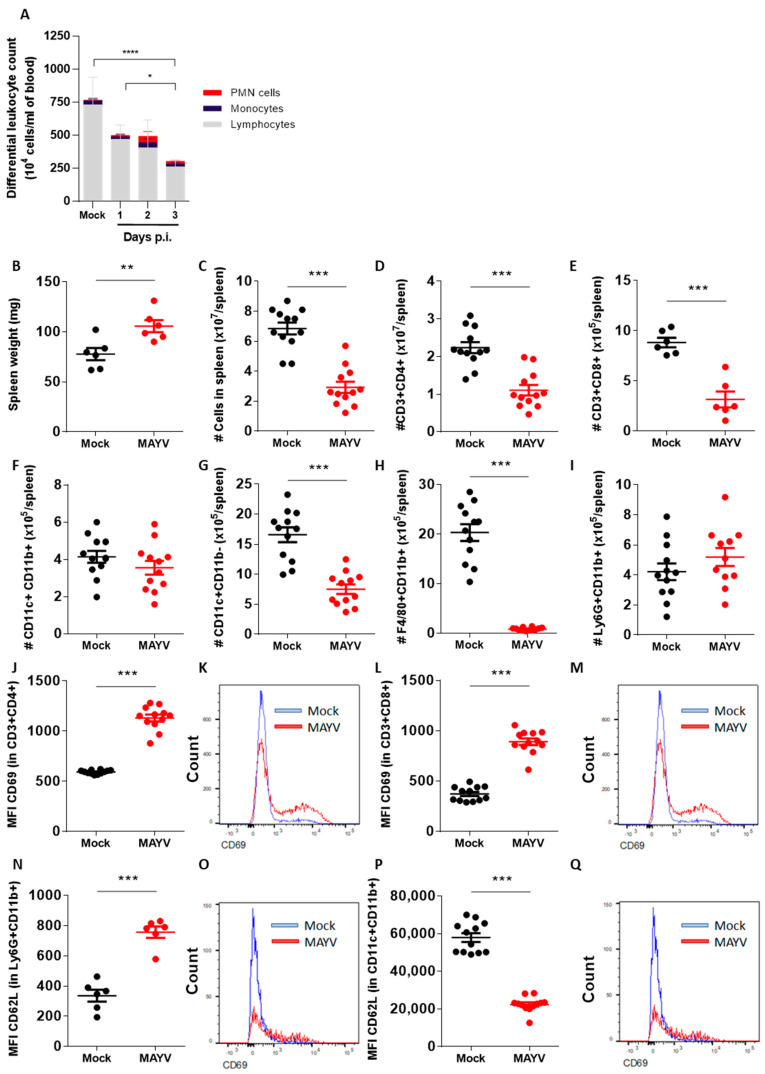
Inoculation of MAYV in immunodeficient IFNAR−/− mice results in lymphopenia and in significant reduction of leukocyte populations in the spleen. (**A**) Differential leukocyte counts from peripheral blood of IFNAR−/− mice infected with 10^2^ PFU MAYV via intraplantar injection at days 1 to 3 p.i. (**B**) Spleen weight of mock- and MAYV-infected mice on day 3 p.i. (**C**–**Q**) Flow cytometry results from spleens of mock- and MAYV-infected mice on day 3 p.i. Absolute numbers of (**C**) total splenic cells, (**D**) T CD4+ and (**E**) T CD8+ lymphocytes, (**F**,**G**) populations of CD11b+ and CD11b- dendritic cells, (**H**) macrophages and (**I**) neutrophils. (**J**) Medium fluorescence intensity (MFI) and (**K**) histogram of CD69 in T CD4+ lymphocytes. (**L**) MFI and (**M**) histogram of CD69 in T CD8+ lymphocytes. (**N**) MFI and (**O**) histogram of CD62L in neutrophils. (**P**) MFI and (**Q**) histogram of MHC-II in macrophages. Data presented as mean ± SEM. * *p* < 0.05, ** *p* < 0.01, *** *p* < 0.001, **** *p* < 0.0001. See also Appendix A.

**Figure 3 ijms-24-04799-f003:**
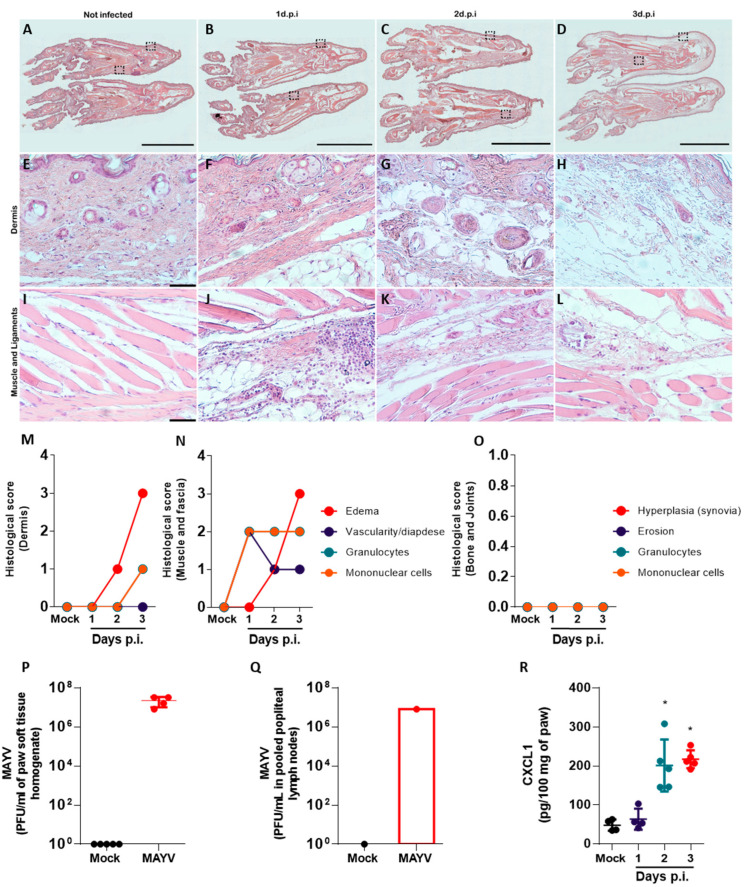
Paw inflammation in MAYV-infected mice involves edema in multiple tissues, local production of chemokine, leukocyte recruitment and infection. Representative photomicrographs of longitudinal footpad sections of mock- and MAYV-infected mice at days 1 to 3 p.i. stained with hematoxylin–eosin, showing (**A**–**D**) overview of the footpad and 20× magnified images of (**E**–**H**) skin and (**I**–**L**) muscles and ligaments. (**M**–**O**) Histological scores were established to evaluate edema, vascularity, diapedesis and cell infiltration in (**M**) skin, (**N**) muscle and ligaments, and (**O**) bone and joint in the footpad sections. MAYV viral load in (**P**) hind paws and (**Q**) in popliteal lymph nodes, and (**R**) CXCL1 levels in hind paws were also assessed in mock- and MAYV-infected mice at day 3 p.i. Data presented as mean ± SEM. * *p* ≤ 0.05. Scale bar (**A**–**D**) 5 mm; (**E**–**L**) 50 µm.

**Figure 4 ijms-24-04799-f004:**
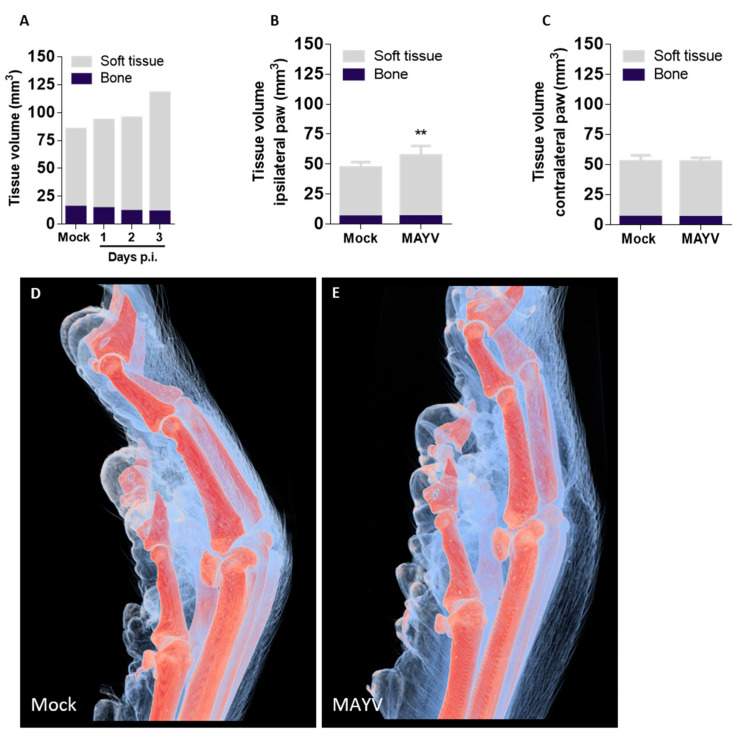
Microtomographic analysis allows precise quantification and localization of edema in paws of MAYV-infected mice in 3D. Three-dimensional reconstructions of mice hind paws were segmented into soft tissue and bone, and tissue volumes were calculated. (**A**) Tissue volumes of mock- and MAYV-infected mice paws at days 1 to 3 p.i. showing increased edema in soft tissues throughout infection. Volumes of soft tissue and bone at day 3 p.i. were also measured in (**B**) ipsilateral (injected) and (**C**) contralateral (non-injected) paws in the same mouse. (**D**,**E**) Visualization of 3D reconstructed paws of (**D**) mock- and (**E**) MAYV-infected mice with segmented soft tissue (blue) and bone (orange). Edema is visible in the MAYV-infected paw, where we observe swollen soft tissue compared to the mock control. Data presented as mean ± SEM. ** *p* <= 0.01.

## Data Availability

The data presented in this study are available on request from the corresponding author.

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
