# Peer review of "Characterization of Systemic Disease Development and Paw Inflammation in a Susceptible Mouse Model of Mayaro Virus Infection and Validation Using X-ray Synchrotron Microtomography"

_ijms, 2023, doi:10.3390/ijms24054799_

Round 1
Reviewer 1 Report
Ana Carolina de Carvalho et al., detailed features of MAYV-induced systemic disease and manifestation of paw edema in a mouse model extensively used in the study of alphavirus disease. The participation of lymphocytes and neutrophils and expression of CXCL1 are key features in both systemic and local manifestations of MAYV disease. The design of this manuscript is reasonable, the results are reliable, the data are reliable, and it has certain guiding significance for the study of Mayaro virus. It is recommended to accept after minor revision.
1. Line 425:5% CO2 ....to 5% CO2....
2. Antibodies and reagents are written in a separate paragraph. “monoclonal antibodies anti-CD3 APC (17A2), CD4 PE (GK1.5), CD8 FITC (53-6.7), CD11b PE-Cy7 (M1/70), CD11c APC-Cy7 (N418), CD25 APC-Cy7 (PC61), CD62L FITC (MEL-14), CD69 PE-Cy7 (H1.2F3), F4/80 PE (BM8), Ly6G APC (1A8), IA/IE Percp-Cy5.5 (M5/114.15.2), ” ect.
3. line 158-162, 211-215, 259-263, 334-339: move to discussion
4. The conclusion should be refined
5. line 591: P<0.05....to p<0.05....
6. line 248: p<=0.05..to p ≤0.05....
Author Response
Reviewer 1
Ana Carolina de Carvalho et al., detailed features of MAYV-induced systemic disease and manifestation of paw edema in a mouse model extensively used in the study of alphavirus disease. The participation of lymphocytes and neutrophils and expression of CXCL1 are key features in both systemic and local manifestations of MAYV disease. The design of this manuscript is reasonable, the results are reliable, the data are reliable, and it has certain guiding significance for the study of Mayaro virus. It is recommended to accept after minor revision.
R: We appreciate the positive comments from reviewer 1.
- Line 425:5% CO2 ....to 5% CO2....
R: We corrected the method in line 480.
- Antibodies and reagents are written in a separate paragraph. “monoclonal antibodies anti-CD3 APC (17A2), CD4 PE (GK1.5), CD8 FITC (53-6.7), CD11b PE-Cy7 (M1/70), CD11c APC-Cy7 (N418), CD25 APC-Cy7 (PC61), CD62L FITC (MEL-14), CD69 PE-Cy7 (H1.2F3), F4/80 PE (BM8), Ly6G APC (1A8), IA/IE Percp-Cy5.5 (M5/114.15.2), ” ect.
R: This information was placed in a separate paragraph.
- line 158-162, 211-215, 259-263, 334-339: move to discussion
R: We appreciate the suggestion, but those segments are merely a summary of the results described in each subsection in Results. Our Discussion section already includes our considerations on each set of results, addressed in seven paragraphs in lines 346 to 472.
- The conclusion should be refined
R: We included concluding sentences at the end of the discussion section, in lines 458 to 472.
- line 591: P<0.05....to p<0.05....
R: Corrected as indicated by the reviewer.
- line 248: p<=0.05..to p ≤0.05....
R: Corrected as indicated by the reviewer.

Reviewer 2 Report
1. At Figure-1 , Figure legend, line-4, the authors need to make superscript for viral load 102 PFU. Please revise it.
2. Regarding the MAYV strain, the authors described the strain was isolated from symptomatic patients. Is there any information about encephalitis or meningoencephalitis? If the authors had information please add it as the viral load levels were highest in brain according to your results.
3. Regarding the viremia level measurements, the authors measured from liver, spleen, brain and footpad only. Why the authors not make measurement from other organs such as muscle, heart, kidney etc? Is there any justification?
4.In this study, did the authors make observation about the body weight, food and water intake of the mice? If you did observation, please add at the revised manuscript.
5.This is the first study about MAYV? The authors used only one strain in this study. Is there any differences between prototype strain and different viral strains from different symtoms patients?
Author Response
Reviewer 2
- At Figure-1 , Figure legend, line-4, the authors need to make superscript for viral load 102 PFU. Please revise it.
R: Corrected as requested.
- Regarding the MAYV strain, the authors described the strain was isolated from symptomatic patients. Is there any information about encephalitis or meningoencephalitis? If the authors had information please add it as the viral load levels were highest in brain according to your results.
R: The patient from which the MAYV strain used in this study (IQT 4235) was isolated presented unspecific febrile illness, which is a common disease symptom arising from MAYV infection in humans. To our knowledge, there are no cases of encephalitis or meningoencephalitis associated with MAYV infection. Headache, retro-orbital pain, and depression are described as possible CNS-related symptoms of arthritogenic alphavirus infection, as reviewed by Levi & Vignuzzi (“Arthritogenic Alphaviruses: A Worldwide Emerging Threat?”; 2019). We interpret the accumulation of MAYV in the brains of IFNAR-/- as a possible limitation of our in vivo infection model, arising from the severe susceptibility of the IFNAR-/- mouse strain to alphavirus infection. Such a condition results in the widespread dissemination of MAYV to mouse tissues, some of which may not recapitulate disease in humans. This consideration was included in the discussion section, lines 369 to 374.
- Regarding the viremia level measurements, the authors measured from liver, spleen, brain and footpad only. Why the authors not make measurement from other organs such as muscle, heart, kidney etc? Is there any justification?
R: Besides from liver, spleen, brain, and paw soft tissue (mostly muscle and ligaments), we also measured viremia in serum and viral load in draining popliteal lymph nodes of mice. Based on the literature, we chose these organs for their importance as target tissues in alphavirus infection, aiming for an overview of MAYV replication and dissemination, and its association with tissue damage and disease signs. Although the addition of heart and kidney samples would be interesting, we found little evidence suggesting the involvement of the heart or the urinary system in arthritogenic alphaviral diseases at the time we conceived the study. Thus, we prioritized an in-depth study of a reduced number of organs.
4.In this study, did the authors make observation about the body weight, food and water intake of the mice? If you did observation, please add at the revised manuscript.
R: Food and water were available ad libitum throughout the experiments, as indicated in the methods section, line 486. We assessed the body weight throughout the infection, as indicated in the results section, lines 110-111, and the graphs can be found in Supplementary Figure 1.
5.This is the first study about MAYV? The authors used only one strain in this study. Is there any differences between prototype strain and different viral strains from different symtoms patients?
R: This is not the first in vivo study about MAYV infection. Other studies of MAYV disease pathogenesis and mechanisms include Figueiredo et al. (2019), Rafael et al. (2020), Santos et al. (2019), Cavalheiro et al. (2016), de Castro-Jorge et al. (2019), all cited and discussed in the manuscript. So far, no comparative in vivo studies using MAYV strains isolated from different symptomatic patients were performed, and some viral strains were not obtained from patients, but from mosquito pools. Distinct in vivo studies using various MAYV strains, all using susceptible IFNAR-/- mice as hosts, indicate similar aspects of disease development, viral replication and dissemination, and describe the involvement of leukocyte populations and expression of cytokines. Figueiredo et al. (2019) describe that MAYV replication and induced inflammation are dependent on age, type-I interferon response and adaptive immunity in mice, which indicates that differences in MAYV disease pathogenesis are mostly dependent on the host’s status than intrinsic viral features. Notwithstanding, Powers et al. (2006) performed molecular genetic analysis of existing MAYV isolates, finding a close genetic relatedness between MAYV strains (less than 15% divergent). In summary, we would not expect major differences in disease pathogenesis when changing the MAYV strain used to infect susceptible IFNAR-/- mice.

Round 2
Reviewer 2 Report
I have no more comments. I accepted in present form.